# Promoter Specificity and Transcription Levels Modulate *Trans*-Splicing Efficiency at the *mod*(*mdg4*) Locus in *Drosophila*

**DOI:** 10.3390/ijms262311609

**Published:** 2025-11-29

**Authors:** Oguljan Beginyazova, Iuliia V. Soldatova, Pavel Georgiev, Maxim Tikhonov

**Affiliations:** Department of Regulation of Genetic Processes, Institute of Gene Biology, Russian Academy of Sciences, 34/5 Vavilov St., 119334 Moscow, Russia; oguljan.1994@mail.ru (O.B.); nao.jem@gmail.com (I.V.S.)

**Keywords:** alternative splicing, *trans*-splicing, transcription factors, *mod*(*mdg4*), promoter, exon, intron

## Abstract

Unlike canonical *cis*-splicing, *trans*-splicing combines exons from two distinct transcripts, creating chimeric mRNAs. One striking example is the *mod*(*mdg4*) locus in *Drosophila*, where all mRNAs, encompassing over 30 isoforms, are exclusively generated via *trans*-splicing, which integrates common constitutive exons with one of the alternative 3’ exons transcribed from independent promoters. This study analyzed the roles of the promoter, exons, and introns in *trans*-splicing, including in the constitutive part of the *mod*(*mdg4*) gene, using a previously validated model in the heterogeneous 22A genomic locus. We found that *trans*-splicing efficiency was significantly greater with the *mod*(*mdg4*) promoter compared to unrelated promoters with the similar transcription level. *Trans*-splicing efficiency correlates positively with transcriptional strength only at low transcription levels, and it does not change with further increases in *mod*(*mdg4*) transcription in the model system. The last common exon and the preceding intron of *mod*(*mdg4*) also play a minor role in enhancing trans-splicing. However, low levels of *trans*-splicing are maintained even when all sequences of the *mod*(*mdg4*) gene are replaced, except for the proximal part of the last intron, where *trans*-splicing is initiated.

## 1. Introduction

During mRNA maturation, eukaryotic pre-mRNAs undergo splicing to remove introns and sequentially join exons [1]. This process is catalyzed by the spliceosome, a dynamic ribonucleoprotein complex comprising five non-coding U-rich small nuclear RNAs and over 170 associated proteins [2,3,4]. Alternative splicing frequently generates multiple mRNA variants from a single pre-mRNA through the use of different splice sites [5,6]. A less common mechanism is *trans*-splicing, which combines exons from two or more distinct pre-mRNA transcripts, creating chimeric with a sequence diversity greater than that produced by canonical *cis*-splicing [7,8]. *Trans*-splicing is generally suppressed in animals, including *Drosophila*, to prevent the formation of potentially dysfunctional chimeric proteins with altered functions, localization, and tissue specificity; however, it may contribute to proteome diversity [9].

The *modifier of mdg4* (*mod*(*mdg4*)) locus in *Drosophila melanogaster* serves as a prime model for *trans*-splicing investigation, as all its over 30 mRNA isoforms arise exclusively through this process (Figure 1A) [10,11,12,13,14,15]. An upstream “donor” segment, common to all mRNA variants, undergoes *cis*-splicing to unite four constitutive exons. These exons are *trans*-spliced to one of multiple “acceptor” exons (Figure 1B). These alternative acceptor exons are transcribed from distinct promoters and are partially situated on opposing DNA strands. In contrast to the alternative *cis*-splicing, *trans*-splicing in *mod*(*mdg4*) amplifies isoform diversity by integrating exons from physically disparate transcripts. *Trans*-splicing also has been shown to be involved in the formation of the large diversity of mRNAs in the *lola* locus [16]. In total, approximately 80 genes have been identified in *Drosophila* for which some mRNA variants can be formed as a result of *trans*-splicing [17].

The existence of the atypical *trans*-splicing process raises fundamental questions about its underlying mechanisms and the factors that govern the switch from canonical *cis*-processing to the *trans*-splicing pathway. The most plausible explanation appears to involve a specific assembly of protein factors that are selectively recruited to the locus, potentially in response to regulatory elements within the promoter, exons, and introns. Despite enormous progress in understanding the regulation of splicing and polyadenylation processes, the mechanisms by which specific *trans*-acting RNA-binding proteins are recruited to genes under their control remain unresolved. Pioneering studies conducted in the late 1990s using transfected minigene reporter constructs showed that the type of promoter used to drive transcription by RNA polymerase II can influence the level of alternative splicing of the downstream exon [18,19]. Experimental results now suggest that various DNA-binding transcription factors, including hormones, may be involved in the regulation of alternative splicing [20,21]. Transcription factors can regulate the transcription elongation rate and participate in the recruitment of specific regulators of alternative splicing to RNA polymerase II. However, the mechanisms underlying these processes remain unknown. Several studies have shown that promoters [22,23] or enhancers [24] determine the specificity of splice site or polyadenylation signal selection.

The selection of alternative splice sites in co-transcriptional alternative splicing is primarily influenced by the elongation rate of RNA polymerase II [25,26]. Significantly, RNA polymerase II exhibits slower elongation at alternatively spliced exons, leading to its accumulation at these genomic regions [27,28,29]. The C-terminal domain (CTD) of the largest subunit of RNA polymerase II—composed of tandem repeats of a conserved heptapeptide sequence—is crucial for splicing and the 3’ end maturation of mRNAs [30]. This CTD also serves as a scaffold for the co-transcriptional recruitment and assembly of RNA-binding proteins involved in alternative splicing and polyadenylation regulation. Any of these factors can directly interact with RNA polymerase II, facilitating rapid recognition of splicing regulatory motifs within the nascent transcript [30,31,32]. Thus, splicing factors can be recruited to upstream regions, travel with RNA polymerase, and, in a specific context, execute a particular splicing program.

The unique splicing strategy, which integrates canonical *cis*-splicing of its initial four exons with the exclusive *trans*-splicing of its alternative 3’ terminal exons, renders the *mod*(*mdg4*) locus an invaluable model for investigating the molecular mechanisms underlying *trans*-splicing. Mechanistically, the *mod*(*mdg4*) gene’s fifth variable exon is likely to be incorporated during mRNA maturation via *trans*-splicing through the recruitment of a specialized *trans*-splicing complex to regulatory sequences. This hypothesis aligns with prior research indicating that regulators of alternative splicing and polyadenylation selectively bind to promoters [33,34,35,36,37], introns [38], and exons [39]. Additionally, particular sequences within intron 4 of *mod*(*mdg4*) have been shown to considerably facilitate *trans*-splicing [14,15]. Our primary objective in this study was to evaluate a model proposing selective recruitment of this *trans*-splicing complex to the promoter and the gene body of *mod*(*mdg4*). This model posits that the complex associates with elongating RNA polymerase II and recognizes particular motifs in the fourth intron, thus initiating *trans*-splicing.

This study investigated the DNA sequences that govern *trans*-splicing at the *mod*(*mdg4*) locus, utilizing a previously established *trans*-splicing model system [15] alongside the φC31 integrase system [40]. Our findings reveal that the *mod*(*mdg4*) promoter enhances *trans*-splicing efficiency several-fold compared to other promoters with similar transcription levels. However, even replacing all *mod*(*mdg4*) sequences up to intron 4 does not completely suppress *trans*-splicing.

## 2. Results

### 2.1. The Model System for Testing the Role of mod(mdg4) Sequences in Trans-Splicing

We investigated the contribution of the *mod*(*mdg4*) promoter and the constant region of the *mod*(*mdg4*) gene to *trans*-splicing using a previously established model [15]. This model involves inserting fragments of the *mod*(*mdg4*) locus into the 22A locus, which is characterized by low transcriptional activity, at a site approximately 15 kb from the nearest gene (Figure 2A). The model comprises two transgenes, a donor and an acceptor, which were both designed to produce their respective transcripts (Figure 2B). Specifically, the donor transgene incorporates a promoter, the gene body, and intron 4. Exon 4 was modified by inserting the internal ribosome entry site (IRES) sequence from the *reaper* (*rpr*) gene near its 3’ end [41]. This sequence was previously used to verify *trans*-splicing at the protein level by initiating luciferase translation [15]; however, in this model, the unique sequence served as a specific target for reverse transcription quantitative PCR (RT-qPCR) analysis. The acceptor transgene featured a bidirectional promoter located between the 3’ exons encoding the T and K isoforms (Figure 1A). The coding regions of the T and K isoforms were replaced with Fluc and mCherry reporters. Simian virus 40 (SV40) polyadenylation signals were inserted at the 3’ ends of the reporters. Strong cap-dependent translation from the acceptor RNA of the two reporters does not allow detection of additional IRES-dependent contribution of the chimeric products. To confirm the obtained results at the protein level, we used a previously developed model system (Appendix A) for some donor variants carrying the *mod*(*mdg4*) promoter.

Site-specific integration of the engineered donor and acceptor constructs into the 22A locus (Figure 2C) was achieved, and their presence in the resulting transgenic lines was confirmed by PCR analysis. After establishing homozygous lines for both donor and acceptor transgenes, lines carrying the acceptor transgene were crossed with various lines containing donor transgenes. This approach exploits the somatic pairing of homologous chromosomes in *Drosophila* [42], which physically juxtaposes both transgenes in the progeny, enabling one allele to function as the donor and the other as the acceptor. Pre-mRNAs are produced from these juxtaposed transgenes, which then undergo *trans*-splicing to generate mature mRNAs.

Total RNA was extracted from 2–3-day-old imago males, and *trans*-splicing events were quantified via RT-qPCR using a forward primer specific to the donor transgene, annealing to the 3’ end of exon 4, a reverse primer complementary to the marker gene within the acceptor transgene, and a TaqMan probe designed to span the splice junction. Separately, the transcription level of the donor gene was quantified using a primer pair comprising a forward primer targeting exon 4 and a reverse primer targeting the IRES sequence, ensuring the PCR product was distinct from the endogenous donor transcript. All transcript levels were normalized to those of the housekeeping genes *vacuolar H^+^ ATPase 100kD subunit 1* (*vha100-1*) and *trafficking protein particle complex subunit 2L* (*CG9067*).

### 2.2. Testing the Functional Role of the mod(mdg4) Promoter in Trans-Splicing

To understand the role of the *mod*(*mdg4*) promoter in *trans*-splicing, we generated a set of transgenic lines in which the native promoter of the *mod*(*mdg4*) gene was replaced with unrelated promoters. As a positive control, we employed a previously characterized transgenic line [15] harboring the donor construct (M>MmMmMmM, four exons [M] and three introns [m] under the control of the native promoter [M>]) from the *mod*(*mdg4*) gene, which includes its promoter, constant region, and intron 4. Conversely, a negative control line used a donor construct in which intron 4, which is crucial for *trans*-splicing, was deleted from the constant region. The *trans*-splicing efficiency of the positive control line was established as a baseline value of 1.

In addition to the *mod*(*mdg4*) promoter (M>, used as our positive control), we selected the strong ubiquitous *actin 5c* (*Act5C*) gene promoter (A>) and the artificial upstream activating sequence (UAS) promoter (U>). The UAS promoter consists of five galactose-responsive transcription factor GAL4 (GAL4) binding sites and the minimal *hsp70Bb* promoter, which acts as a weak ubiquitous promoter in the absence of the yeast GAL4 activator.

We first evaluated *trans*-splicing efficiency using donor transgenes containing the *mod*(*mdg4*) gene segment (MmMmMmM) under the control of various promoters (Figure 3). Although the *mod*(*mdg4*) and *Act5C* promoters generated similar levels of transcription, *trans*-splicing efficiency was approximately halved when the *Act5C* promoter was utilized. In contrast, while the UAS promoter drove considerably lower transcription than the *Act5C* promoter, the resulting *trans*-splicing efficiency remained comparable to that achieved with the *Act5C* promoter.

We next investigated the role of introns in the *mod*(*mdg4*) gene in *trans*-splicing. For this purpose, we generated a series of donor transgenes in which the *mod*(*mdg4*) gene was replaced with the cDNA (MMMM, Figure 3). Transcription of the MMMM construct was 3- to 4-fold lower when driven by the *Act5C* and *mod*(*mdg4*) promoters (Figure 3). In contrast, its transcription was at least 10-fold lower when driven by the UAS promoter. The lower transcription with the cDNA can be explained by both the decreased stability of intronless transcripts and the presence of enhancers in the introns of the *mod*(*mdg4*) gene.

The difference in *trans*-splicing levels between the *mod*(*mdg4*) and *Act5C* promoters, and particularly the UAS promoter, became considerably greater when using the MMMM construct. *Trans*-splicing efficiency was 50% lower with the MMMM construct than with the MmMmMmM construct, which can be explained, at least partly, by the significantly lower transcription with the MMMM construct than with the MmMmMmM construct. The reduction in *trans*-splicing with the MMMM construct was 2–4-fold greater when driven by the *Act5C* promoter than by the *mod*(*mdg4*) promoter, confirming the role of the *mod*(*mdg4*) promoter in maintaining efficient *trans*-splicing. Moreover, despite a 10-fold difference in transcription levels, the *Act5C* and UAS promoters maintained approximately the same level of *trans*-splicing.

### 2.3. Testing the Functional Role of Intron 3 and Exon 4 in mod(mdg4) in Trans-Splicing

To identify elements within the constant region of *mod*(*mdg4*) influencing *trans*-splicing, we substituted segments in the MmMmMmM construct with those from the unrelated *glutamate receptor-interacting protein* (*grip*) gene. We chose *grip* because its exon and intron sizes are similar to those in the constant region of *mod*(*mdg4*), and it does not exhibit complex alternative splicing (Figure 4A). We generated a series of three constructs (Figure 4B) based on *grip* sequences (designated GgGgGgG, consisting of four exons [G] and three introns [g]). In the GgGgGgG construct, the constant region of the *mod*(*mdg4*) gene was substituted entirely by the *grip* gene. Then, exon 4 or intron 3 of the donor gene *mod*(*mdg4*) was incorporated into this construct, designated GgGgGgM and GgGgGmG, respectively. All these constructs were combined with one of the tested promoters: *mod*(*mdg4*), *Act5C*, or UAS.

Again, we found that transcription of the constructs was markedly reduced, confirming that an enhancer is located in at least one of the *mod*(*mdg4*) introns (Figure 4B). The results with all tested constructs also confirmed that the *mod*(*mdg4*) promoter is much more efficient in *trans*-splicing than the UAS and *Act5C* promoters. Replacing exon 4 or intron 3 of the *grip* gene with the corresponding *mod*(*mdg4*) sequences resulted in a slight increase in *trans*-splicing. Thus, these sequences appear to have only a minor role in *trans*-splicing.

Replacement of the first three exons and two introns of the *grip* gene with the corresponding *mod*(*mdg4*) sequences (MmMmMgG) markedly increased transcription from the *mod*(*mdg4*) promoter (Figure 4B). Thus, the enhancer is localized in intron 1 or 2 of the *mod*(*mdg4*) gene. Despite the increased transcription, this substitution did not affect *trans*-splicing efficiency. Again, the additional substitution of *grip* intron 3 by *mod*(*mdg4*) (MmMmMmG) intron 3 had a weak positive effect on *trans*-splicing.

To further validate our findings, we performed an analysis using a model system to assess *trans*-splicing efficiency at the protein level (Appendix A). We evaluated the activity of Fluc, which could only be translated from chimeric transcripts. *Trans*-splicing efficiencies were comparable for the GgGgGgM and MmMmMmM donor transgenes. In the other donor transgenes studied (GgGgGgG, GgGgGmG, MmMmMgG, MmMmMmG), *trans*-splicing activity was reduced (Appendix A). Thus, a significant correlation was obtained between the results obtained by the two methods.

### 2.4. Sequences in the Proximal Part of Intron 4 in mod(mdg4) are Sufficient to Induce Trans-Splicing

The results demonstrated that replacing the promoter, constitutive exons, and introns of the *mod*(*mdg4*) gene reduced *trans*-splicing but did not completely inactivate it. Thus, sequences in the proximal part of intron 4 appear sufficient to induce *trans*-splicing. To confirm this hypothesis, we examined the cDNA of the *grip* gene (GGGG) under the control of the *mod(mdg4) (M>)*, *Act5C* (A>), and UAS (U>) promoters (Figure 4C). As expected, transcription of the M>GGGG construct was low, amounting to 30% of the control construct. However, unexpectedly, the T isoform achieved *trans*-splicing equivalent to the control construct, while *trans*-splicing of the K isoform was 40% of the control construct. Substitution of the *mod(mdg4) promoter* with the *Act5C* or UAS promoter resulted in a 4- and 20-fold decrease in *trans*-splicing, respectively. Nevertheless, *trans*-splicing was maintained even when using a donor containing the GGGG construct under the control of the UAS or *Act5C* promoter; in this case, the entire region upstream of intron 4 is replaced with heterologous sequences. This result suggests a key role for the proximal part of the last common intron 4 in *trans*-splicing.

### 2.5. Correlation Between Transcription of the Donor and Trans-Splicing Level

We used the varying transcriptional levels observed among the generated constructs to investigate the correlation between donor gene transcription and *trans*-splicing efficiency. A comparison using the *Act5C* and UAS promoters revealed a strong positive correlation between transcription levels and *trans*-splicing efficiency, particularly at lower transcription levels (up to 0.4 relative to the control construct, Figure 5A). At low to moderate expression levels of donor mRNA, *trans*-splicing efficiency exhibited a proportional increase (R^2^ = 0.81). This finding suggests that at low and moderate transcription levels, *trans*-splicing efficiency is primarily determined by transcriptional strength. In contrast, constructs incorporating the putative enhancer (A>MmMmMmM and A>MmMmMmG) and exhibiting high transcription levels reached a maximum *trans*-splicing efficiency of only approximately 60% relative to the control construct. This observation suggests that, under strong transcription, *trans*-splicing efficiency does not directly correlate with transcription levels. A discernible upper limit is observed in the quantity of *trans*-spliced products.

To validate this hypothesis, we increased transcription under the UAS promoter by introducing the *Act5C* promoter-controlled *gal4* transgene into the U>MmMmMmM line (Figure 5B). As expected, GAL4 bound to its recognition sites within the UAS promoter and activated transcription, increasing transcription levels by approximately 30-fold. However, *trans*-splicing efficiency only increased modestly by 1.5–2-fold, reaching values comparable to those observed in the control line (a relative efficiency of approximately 1.0). These findings confirm that, at high transcription levels, no direct correlation exists between transcription levels and *trans*-splicing efficiency. The quantity of *trans*-splicing products reaches a plateau, which may be attributable to limitations in the acceptor pre-mRNA pool or the *trans*-splicing factors facilitating the process.

## 3. Discussion

As all *mod*(*mdg4*) mRNAs are generated exclusively via *trans*-splicing [15,43], one obvious explanation was that the *mod*(*mdg4*) regulatory sequences might specifically recruit a protein complex that initiates *trans*-splicing. However, our results show that the proximal sequences of intron 4 alone are sufficient to induce *trans*-splicing with low efficiency. Thus, no specific complex is recruited exclusively to the *mod*(*mdg4*) promoter or gene body and determines *trans*-splicing.

An enhancer appears to be present in the proximal part of the *mod*(*mdg4*) gene body, most likely in the first intron, which stimulates the activity of the *mod*(*mdg4*) and *Act5C* promoters. Both promoters can be classified as strong ubiquitous promoters controlled by nearby proximal enhancers. The third promoter used in our experiments was artificial and consisted of a minimal *hsp70Bb* promoter linked to five binding sites for the yeast activator GAL4. The presence of the *mod*(*mdg4*) gene enhancer markedly increased expression driven by the UAS core promoter. However, even in the absence of the *mod*(*mdg4*) enhancer, weak transcription driven by the UAS core promoter was sufficient for *trans*-splicing in the model system. Interestingly, induction of the UAS promoter by GAL4 resulted in a marked increase in transcription but only a modest increase in *trans*-splicing. Thus, there is no direct correlation between the transcription levels of the 5’ donor part of *mod*(*mdg4*) and *trans*-splicing efficiency. In the model system, this could be explained by the relatively weak transcription of acceptor exons under the examined promoters or by limited number of unknown factors required for *trans*-splicing.

*Cis*-splicing of most metazoan pre-mRNAs is generally regarded as independent of promoter and expression levels, although several studies highlight notable exceptions. For instance, it has been demonstrated that NSP1 pre-mRNA shows promoter-specific and transcription rate-dependent variations in splicing efficiency, suggesting that promoter architecture and transcription kinetics can influence spliceosome activity [44]. Recent research has revealed that the utilization of alternative transcription initiation sites represents a widespread regulatory mechanism that directly impacts mRNA isoform formation. The use of different transcriptional start sites (TSS) can determine which splicing and polyadenylation sites are employed, thereby linking promoter architecture to posttranscriptional RNA processing [22,23].

Overall, our results show that the *mod*(*mdg4*) promoter markedly increases *trans*-splicing, whereas the *mod*(*mdg4*) gene body has only a minor positive effect. Therefore, it is conceivable that the *mod*(*mdg4*) promoter and elements within its gene body (specifically intron 3 and exon 4) recruit certain transcription factors and splicing regulators to facilitate *trans*-splicing. For instance, the embryonic lethal abnormal vision (ELAV) protein binds to polyadenylation signaling motifs within neuronal genes to promote the elongation of their 3’-UTRs [45]. ELAV has been demonstrated to be selectively recruited to promoters and introns to mediate 3’-UTR extension in neuronal cells [35]. Analogously, the transcriptional coactivator *nejire* (*nej*, formerly called p300/CBP) modulates splicing and polyadenylation upon its recruitment to promoters [23]. Notably, brahma (brm), the ATPase subunit of the SWI/SNF chromatin-remodeling complex, has been previously shown to positively regulate *trans*-splicing at the *mod*(*mdg4*) locus [13]. Therefore, it is plausible that brm is more efficiently recruited to the *mod*(*mdg4*) promoter and gene body compared to the other promoters examined. The *mod*(*mdg4*) promoter can also more efficiently recruit specific RNA-binding proteins to RNA polymerase II, which then participate in *trans*-splicing.

*Trans*-splicing considerably expands the capacity for alternative splicing, enabling a single genomic locus to yield an extensive array of mRNAs encoding diverse protein isoforms. Given that *trans*-splicing relies primarily on intronic sequence motifs [14,15] rather than specialized splicing complexes, it can readily coexist with *cis*-splicing at the same locus, contributing to greater transcriptomic diversity. Consequently, *trans*-splicing may be far more pervasive than currently recognized. Nevertheless, comprehending its full scope and biological importance necessitates unraveling its underlying molecular mechanisms, which currently remain elusive.

## 4. Materials and Methods

### 4.1. Construct Design and Molecular Cloning

Genetic constructs were prepared according to standard molecular biology protocols, utilizing reagents from Thermo Fisher Scientific (Waltham, MA, USA). Plasmids were propagated in *Escherichia coli* strain DH5α under appropriate selective conditions. Site-specific integration constructs were developed from a pBluescript-derived backbone, incorporating *mini-white*, a loxP site, a multiple cloning site, and an attB site for φC31 integrase-mediated insertion. Promoters were acquired from their respective sources: the *mod*(*mdg4*) promoter was PCR-amplified from *D. melanogaster* genomic DNA, the *Act5C* promoter was excised from the pAc5.1/V5-His plasmid, and the UAS promoter was obtained from the pUAST plasmid [46]. Genomic DNA was used as the template for PCR amplification of the exons and introns in *mod*(*mdg4*) and *grip*, and the IRES of *rpr*. Various exon–intron combinations of *mod*(*mdg4*) and *grip* were generated using overlap-extension PCR or classical restriction enzyme cloning. Promoters, outrons, and short 5’ exon fragments for the acceptor constructs were also amplified from genomic DNA. Firefly luciferase (*Fluc*) was excised from the pGL3-Basic plasmid (Promega Corporation, Madison, WI, USA), while mCherry was PCR-amplified from a laboratory stock vector. The SV40 polyadenylation signal was obtained from the pAc5.1/V5-His plasmid and inserted downstream of the marker genes. The oligonucleotides used are listed in Appendix A.

### 4.2. Drosophila Stocks, Transgenesis, and Genetic Crosses

*D. melanogaster* stocks were maintained at 25 °C under standard conditions. Transgenic constructs were integrated site-specifically into the 22A locus using φC31 integrase-mediated recombination. Plasmid DNA (200 ng/μL) was microinjected into preblastoderm embryos [47] from a fly line (stock #: 24481, Bloomington *Drosophila* Stock Center: *y^1^ M{vas-int.Dm}ZH-2A w^1118^; M{3xP3-RFP.attP’}ZH-22A*) [40] that carries an attP docking site and expresses φC31 integrase under the germline-specific *vas* promoter. F0 flies were crossed with *y^1^w^1118^* flies, and F1 progeny with the transgene were selected via *mini-white* eye pigmentation. Homozygous lines were established by crossing with *y^1^w^1118^*; *Kr^If−1^*/SM5 flies. Integration was confirmed by PCR across transgene–genome junctions and sequencing of selected products.

The GAL4 transgene line (stock #: 4414; Bloomington *Drosophila* Stock Center, Bloomington, IN, USA) was recombined with a line carrying the donor construct (U>MmMmMmM). The resulting double-transgene line was crossed with acceptor homozygotes to generate experimental flies.

### 4.3. Quantification of Splicing Efficiency and Gene Expression

Biological replicates, consisting of 15–30 adult males (2–3 days old), were flash-frozen in liquid nitrogen. Next, samples were homogenized in 200 µL of TRIzol Reagent (Medical Research Center, Cincinnati, OH, USA), adjusted to a final volume of 1 mL, and total RNA was extracted according to the manufacturer’s protocol. Then, total RNA was treated with DNase I (Thermo Fisher Scientific) to eliminate residual genomic DNA. Next, cDNA was synthesized using random hexamer primers with the RevertAid Maxima Reverse Transcriptase (Thermo Fisher Scientific) or LunaScript RT Master Mix (Primer-free; New England Biolabs, Ipswich, MA, USA), following the respective manufacturer’s guidelines. RT-qPCR was performed using Diamant TaqA polymerase (BelBioLab, Moscow, Russia) with either EvaGreen dye (Biotium, Inc., Fremont, CA, USA) or TaqMan probes (using ROX as a reference dye; Thermo Fisher Scientific) on a QuantStudio 6 Flex Real-Time PCR System (Thermo Fisher Scientific). Relative mRNA levels were determined through standard curves derived from serial dilutions of genomic DNA (for housekeeping genes) or PCR products spanning splice junctions (for splicing efficiency). Data normality was assessed using the Shapiro–Wilk test, and all datasets passed normality assumptions (*p* > 0.05). Equality of variances was evaluated via F-tests; variances were unequal in some comparisons. Statistical comparisons between experimental and control groups were conducted using Welch’s *t*-test (two-sample *t*-test assuming unequal variances) implemented in Excel via the T.TEST function with two tails and unequal variances. For multiple comparisons, the Bonferroni correction was applied.

### 4.4. Luciferase Analysis

For each replicate, 15 adult 2- to 3-day-old males were collected, frozen in liquid nitrogen, and homogenized in 150 μL of Firefly Lysis Buffer (Biotium). The resulting lysate was centrifuged at 2000× *g* for 5 min, and 20 μL of the clear supernatant was assayed using the Firefly Luciferase Assay Kit (Biotium) according to the manufacturer’s instructions. Luciferase activity was measured using a GloMax 20/20 luminometer (Promega). Total protein concentration in the lysates was quantified fluorometrically using a Qubit fluorometer (Thermo Fisher Scientific). Assays were performed using at least five independent biological replicates.

## Figures and Tables

**Figure 1 ijms-26-11609-f001:**
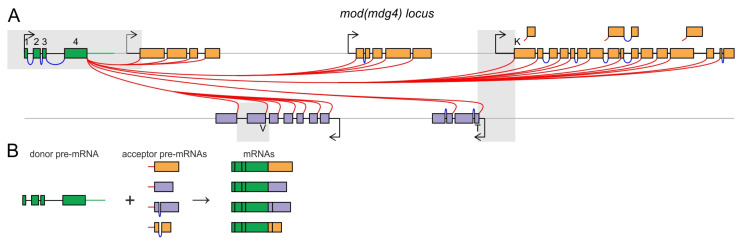
The *mod*(*mdg4*) locus and its associated *trans*-splicing. (**A**) Structure of the *mod*(*mdg4*) locus in *Drosophila melanogaster*. Annotated promoters are indicated by arrows. Constitutive exons, shared across all isoforms, are shown in green and numbered from 1 to 4. Alternative exons are shown in orange when located on the same DNA strand as the common exons, and in purple when on the opposite strand. Constitutive introns are marked by blue arcs. Variants of *trans*-splicing are shown by red curves. The DNA fragments used to generate the donor and acceptor constructs in this study are shown in gray boxes. The alternative exons corresponding to isoforms T, K, V, which were utilized to construct the acceptor transgene, are indicated accordingly. (**B**) Scheme of *trans*-splicing. The donor part of the locus produces the donor pre-mRNA (green). The acceptor part generates multiple acceptor pre-mRNAs (orange and purple). Through an unknown *trans*-splicing mechanism, donor and acceptor pre-mRNAs combine to form 31 mRNA variants. Each mature mRNA consists of five or six exons, four of which derive from the constitutive part.

**Figure 2 ijms-26-11609-f002:**
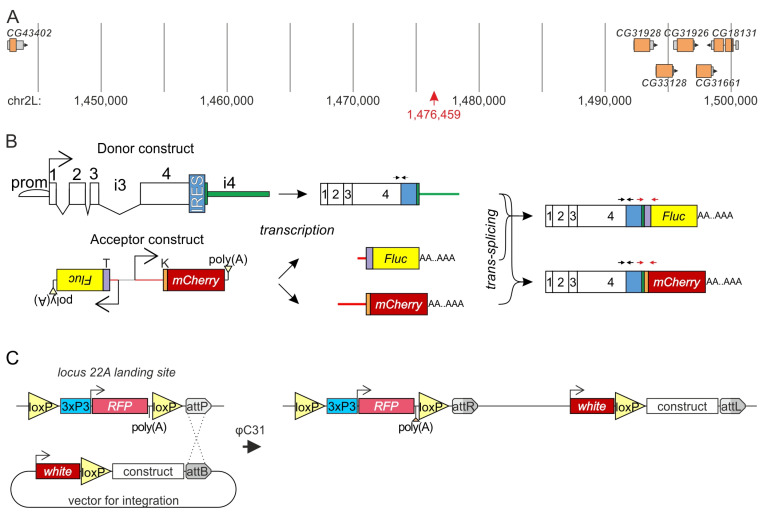
The model system for testing the role of *mod*(*mdg4*) sequences in *trans*-splicing. (**A**) Genomic map of the 22A locus. The red arrow marks the integration site. The orange-colored neighboring genes are more than 15 kb away. The coordinates correspond to chromosome 2L nucleotide numbering. (**B**) Schematic of the model system developed to investigate the functional roles of the *mod*(*mdg4*) gene regions in *trans*-splicing. The donor construct (top) features a promoter (indicated by an arrow and semicircle), constitutive exons (rectangles numbered from 1 to 4), introns (angle brackets; i3 denotes the third intron) an internal ribosome entry site (IRES, blue rectangle) sequence, and intron 4. The acceptor construct (bottom) incorporates bidirectional promoters (arrows) and outrons (red lines) specific to isoforms T and K. Fluc (yellow rectangle) and mCherry (red rectangle) reporters are strategically placed immediately downstream of their respective exon start sites. Transcription yields a single donor transcript and two distinct acceptor transcript variants. Then, *trans*-splicing produces two mRNA variants, each resulting from the fusion of the 5’ donor portion with one of the two 3’ acceptor variants. Above the transcripts, red and black arrows denote primer annealing sites for assessing *trans*-splicing efficiency and donor transcript levels, respectively. (**C**) Schematic of the integration of donor and acceptor model constructs into the 22A locus. The 22A locus is equipped with an attP docking site and a red fluorescent protein (RFP) reporter, driven by the 3 × P3 promoter. The φC31 integrase gene, controlled by the germline-specific *vasa* (*vas*) promoter, is also integrated on the X chromosome [40]. Integration vectors contain donor or acceptor sequences, a *white* reporter, and an attB site. Site-specific recombination between attP and attB sites results in genomic integration of the entire plasmid. Successful integration events were identified by the presence of pigmented eyes.

**Figure 3 ijms-26-11609-f003:**
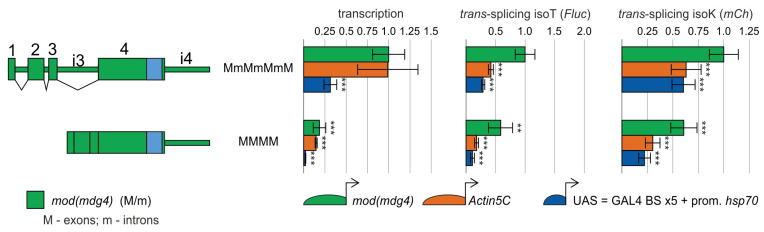
Investigation of the roles of promoter and the first three introns of the *mod*(*mdg4*) gene in *trans*-splicing. Schematic representations of the donor constructs used to investigate the roles of the *mod*(*mdg4*) gene components are shown on the left. Fragments originating from *mod*(*mdg4*) are shown in green, and the IRES is shown in light blue. Construct nomenclature follows this convention: uppercase letters designate exons, lowercase letters denote introns, and M/m represents *mod*(*mdg4*). On the right, transcription levels and *trans*-splicing efficiency in donor/acceptor heterozygotes are presented, quantified by RT-qPCR and evaluated based on the ratio of the corresponding amplicon to two housekeeping genes (*vha100-1* and *CG9067*). Donor constructs were placed under the transcriptional control of the *mod*(*mdg4*) (green histograms), *Act5C* (orange histograms), and UAS (blue histograms) promoters, each indicated by an arrow and semicircle. Each donor/acceptor *trans*-heterozygote was analyzed in at least three replicates. Error bars represent the standard deviations (*n* = 3–13). Asterisks indicate significance: **, *p* < 0.01; ***, *p* < 0.001.

**Figure 4 ijms-26-11609-f004:**
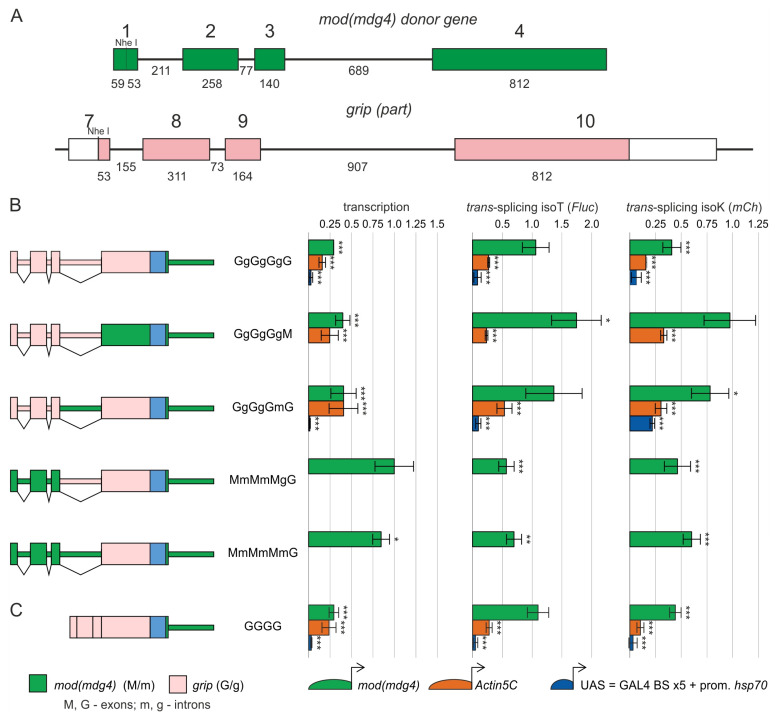
Investigation of the roles of promoter, exons, and introns in *mod(mdg4) trans*-splicing. (**A**) Scaled diagram of a *grip* gene fragment (pink) used to replace sequences of the *mod*(*mdg4*) gene (green). Exon and intron lengths are indicated below the schematic, and exon numbering is relative to the gene start and shown above. (**B**,**C**) Schematic representations of the donor constructs used to investigate the roles of the *mod*(*mdg4*) gene components are shown on the left. Fragments originating from *mod*(*mdg4*) are shown in green, and substitutions from the *grip* gene are shown in pink; the IRES is shown in light blue. Construct nomenclature follows this convention: uppercase letters designate exons, lowercase letters denote introns, M/m represents *mod*(*mdg4*), and G/g represents *grip*. On the right, transcription levels and *trans*-splicing efficiency in donor/acceptor heterozygotes are presented, quantified by RT–qPCR and evaluated based on the ratio of the corresponding amplicon to two housekeeping genes (*vha100-1* and *CG9067*). Donor constructs were placed under the transcriptional control of the *mod*(*mdg4*) (green), *Act5C* (orange), and UAS (blue) promoters. Each donor/acceptor *trans*-heterozygote was analyzed in at least three replicates. Error bars represent the standard deviations (*n* = 3–13). Asterisks indicate significance: *, *p* < 0.05; **, *p* < 0.01; ***, *p* < 0.001.

**Figure 5 ijms-26-11609-f005:**
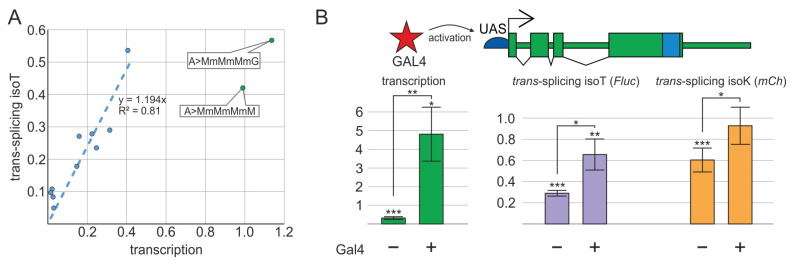
The relationship between transcription levels and *trans*-splicing efficiency. (**A**) A linear relationship between transcription levels and *trans*-splicing efficiency is evident for constructs under the control of the *Act5C* (A>) and UAS (U>) promoters (blue points and line) at low transcription levels. Data for two constructs exhibiting enhanced transcription are also shown (green). (**B**) Assessment of the impact of robust UAS promoter activation by GAL4 on *trans*-splicing. The schematic above shows the GAL4 protein (red asterisk) activating the UAS promoter (blue semicircle), driving transcription of the *mod*(*mdg4*) constitutive region (green). The barplot below shows transcription and *trans*-splicing levels in donor–acceptor heterozygotes with active (+) versus inactive (−) UAS promoters. Each donor/acceptor *trans*-heterozygote was analyzed in three replicates. Error bars represent the standard deviations (*n* = 3). Asterisks indicate significance: *, *p* < 0.05; **, *p* < 0.01; ***, *p* < 0.001.

## Data Availability

Inquiries can be directed to the corresponding author.

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
