# Peer review of "Promoter Specificity and Transcription Levels Modulate Trans-Splicing Efficiency at the mod(mdg4) Locus in Drosophila"

_ijms, 2025, doi:10.3390/ijms262311609_

Round 1
Reviewer 1 Report
Comments and Suggestions for Authors
Beginyazova et al study the quantitative determinants of the efficiency of trans-splicing. They find that splicing efficiency correlates positively with transcriptional strength only at low transcription levels. The findings are interesting however the presentation, especially the introduction should be improved. Furthermore, the quantitation and the interpretation of experiments should be extended.
Major comments
1. Readability:
B. Introduction: The authors introduce details of splicing which is hardly ever used, while the cis-splicing and trans-splicing is not sufficiently contrasted. For example, it is not clear why the authors introduce serine and arginine-rich proteins, even though they are not mentioned later. Furthermore, it is better to avoid cis-elements and trans-acting factors. Per se they are all correct, however, the cis / trans distinction is already used to define the two types of splicing the extra layer of usage of cis/trans regulators is somewhat confusing, especially these definitions are not used later in the text.
B. Introduction / discussion: Important pieces of information are found only in Discussion, such that trans-splicing is generally suppressed; should be moved to the Introduction. Furthermore, trans-splicing should be introduced more thoroughly. For example, their role in chimeras should be discussed. Frenkel-Morgenstern et al “Chimeras taking shape: Potential functions of proteins encoded by chimeric RNA transcripts»
C. Figures should be better labelled and structured. For example, the “…intron 4 is denoted by a dashed line.” However, the dashed line is hardly visible in Fig. 1. The red lines should be also made prominent.
D. Fig. 1 and 2 should be better matched. What are the elements in Fig.1 that were reused in Fig. 2. These elements should be already marked (e.g. with a star) in Fig. 1.
2. mRNA quantification and interpretation of results.
A. The authors hypothesize that trans-splicing becomes saturated at high level. However, saturation can be substantiated only if both donor and acceptor mRNAs are exactly quantified and shown that they reach similar levels close to saturation. This can be done with digital droplet PCR or by carefully measuring the amplification efficiency of the primers and reporting efficiency corrected RNA values for the compared mRNAs.
B. The authors should discuss and compare the expression- and promoter-dependence of the cis- and trans-splicing. The efficiency of cis-splicing of most mRNAs is independent of expression level and promoters. However, there are exceptions, eg. The NSP1 mRNA whose splicing efficiency depends both on promoter identity and transcription rate (Bonde et al: “Quantification of pre-mRNA escape rate and synergy in splicing”).
Author Response
Thank you very much for your thorough and constructive review of our manuscript. Please find our detailed responses below, as well as the revised manuscript with track changes in the resubmitted files.
1. Readability:
A. Introduction: The authors introduce details of splicing which is hardly ever used, while the cis-splicing and trans-splicing is not sufficiently contrasted. For example, it is not clear why the authors introduce serine and arginine-rich proteins, even though they are not mentioned later. Furthermore, it is better to avoid cis-elements and trans-acting factors. Per se they are all correct, however, the cis / trans distinction is already used to define the two types of splicing the extra layer of usage of cis/trans regulators is somewhat confusing, especially these definitions are not used later in the text.
We've rewritten the introduction. We removed mentions of facts not directly relevant to understanding the article, including references to serine- and arginine-rich proteins. We also replaced 'trans-acting factors' with 'RNA-binding proteins' to improve precision.
B. Introduction / discussion: Important pieces of information are found only in Discussion, such that trans-splicing is generally suppressed; should be moved to the Introduction. Furthermore, trans-splicing should be introduced more thoroughly. For example, their role in chimeras should be discussed. Frenkel-Morgenstern et al “Chimeras taking shape: Potential functions of proteins encoded by chimeric RNA transcripts»
We moved the text from the Discussion to the Introduction. We added content on the role of trans-splicing in generating chimeric products and expanded the section dedicated to trans-splicing.
C. Figures should be better labelled and structured. For example, the “…intron 4 is denoted by a dashed line.” However, the dashed line is hardly visible in Fig. 1. The red lines should be also made prominent.
We enlarged the text in the figures that may have been difficult to read. We revised the first figure and increased the contrast across all figures for better visibility.
D. Fig. 1 and 2 should be better matched. What are the elements in Fig.1 that were reused in Fig. 2. These elements should be already marked (e.g. with a star) in Fig. 1.
We highlighted the locus elements used to generate the constructs with gray shading. We also marked the isoforms used for the constructs and emphasized these details in the figure caption.
2. mRNA quantification and interpretation of results.
A. The authors hypothesize that trans-splicing becomes saturated at high level. However, saturation can be substantiated only if both donor and acceptor mRNAs are exactly quantified and shown that they reach similar levels close to saturation. This can be done with digital droplet PCR or by carefully measuring the amplification efficiency of the primers and reporting efficiency corrected RNA values for the compared mRNAs.
We thank the reviewer for bringing this issue to our attention. We agree that our previous formulation of the results discussed in this section was not entirely precise. Our intent was to convey that increasing transcription leads to a rise in the abundance of trans-spliced products (notably, we quantified this with high precision – for each primer pair, we constructed calibration curves using serial dilutions of standards, and normalized the levels of donor construct transcription and trans-splicing extent against two genes exhibiting stable expression). This increase reaches an upper limit that approximates the outcomes observed with the unmodified control construct. We have rewritten this section to provide more precise formulations.
B. The authors should discuss and compare the expression- and promoter-dependence of the cis- and trans-splicing. The efficiency of cis-splicing of most mRNAs is independent of expression level and promoters. However, there are exceptions, eg. The NSP1 mRNA whose splicing efficiency depends both on promoter identity and transcription rate (Bonde et al: “Quantification of pre-mRNA escape rate and synergy in splicing”).
We have added a section to the Discussion dedicated to the efficiency of cis- and trans-splicing depending on the promoter type.
Reviewer 2 Report
Comments and Suggestions for Authors
The manuscript presents a detailed molecular analysis of promoter-dependent modulation of trans-splicing efficiency at the mod(mdg4) locus in Drosophila melanogaster. However, while the findings are interesting, the study remains somewhat descriptive. The manuscript is well-written, but several additional experiments are necessary.
Major Comments
- The conclusions regarding the influence of promoter strength and intronic regions on trans-splicing efficiency are drawn solely from RT-qPCR quantification.
- Consider testing promoter effects using additional unrelated promoters to generalize the findings beyond mod(mdg4), Act5C, and UAS.
- Assess whether altered transcription or splicing efficiency translates into differences in reporter protein levels (luciferase or mCherry).
- The supplementary materials list primer sequences but lack explicit 5′→3′ orientation and full sequence information for some oligonucleotides.
- The discussion claims that the proximal part of intron 4 is sufficient for trans-splicing, but the experimental data do not fully rule out enhancer or chromatin effects. Include a control construct
- Although p-values are shown, there is no indication of the test type, variance assumptions, or multiple-comparison corrections. Specify whether the data passed normality tests and which statistical tests (e.g., Student’s t, ANOVA) were used.
- The study extends previous findings by the same group. The authors should emphasize what new mechanistic insight is gained here.
Minor Comments
- Figures are informative but dense. Label font size is small.
- Include specific concentrations and amounts of plasmid DNA used
- Authors should deposit plasmid maps, construct sequences in a repository (GEO).
Author Response
Thank you very much for your thoughtful and insightful comments on our manuscript. Our point-by-point responses are provided below, and the revised version with track changes is included in the resubmitted files.
“The conclusions regarding the influence of promoter strength and intronic regions on trans-splicing efficiency are drawn solely from RT-qPCR quantification.”
In the Supplementary, we've included results demonstrating trans-splicing at the protein level for some of the tested variants. Based on our data from this and previous work (Tikhonov et al 2018), RT-qPCR provides a more accurate measurement of trans-splicing intensity.
“Consider testing promoter effects using additional unrelated promoters to generalize the findings beyond mod(mdg4), Act5C, and UAS.”
We tested two well-studied heterologous promoters. One promoter is one of the strongest housekeeping gene promoters (Act5C), which is often used for gene expression. The second promoter, UAS, is an artificial promoter consisting of the minimal promoter of the heat shock gene 70 and five sites for the yeast activator GAL4. These promoters were shown to be less efficient than the mod(mdg4) promoter but still support trans-splicing. Investigating additional promoters would be useful if the properties of such promoters were well characterized. We are not aware of any such promoters.
“Assess whether altered transcription or splicing efficiency translates into differences in reporter protein levels (luciferase or mCherry)”
At the reviewer's request, we have added some results of the trans-splicing study at the protein level to the Supplementary section. A direct correlation is observed between the results obtained in the trans-splicing studies at RNA- and protein- levels.
“The supplementary materials list primer sequences but lack explicit 5′→3′ orientation and full sequence information for some oligonucleotides.”
In accordance with the reviewer's request, we have modified the table with primers.
“The discussion claims that the proximal part of intron 4 is sufficient for trans-splicing, but the experimental data do not fully rule out enhancer or chromatin effects. Include a control construct”
The question is whether intron 4 itself can induce trans-splicing. Initially, we also believed that trans-splicing requires the entire context: the promoter and the gene body with exons and introns. However, in this work, we made a very interesting observation that if we take a different promoter (Act5C or UAS) and the gene body from grip, trans-splicing is preserved. In other words, the presence of the fourth intron alone is sufficient to maintain this process.
Regarding chromatin, we placed the transgene in a different locus, where the chromatin context likely differs from that of the endogenous mod(mdg4). However, trans-splicing was preserved. Therefore, the most plausible assumption is that the sequences from intron 4 are sufficient for trans-splicing, while the promoter and gene body may contribute to its efficiency.
“Although p-values are shown, there is no indication of the test type, variance assumptions, or multiple-comparison corrections. Specify whether the data passed normality tests and which statistical tests (e.g., Student’s t, ANOVA) were used.”
We thank the reviewer for this comment. We made a mistake by not describing this section in detail. We have corrected this and added the text to the "Materials and Methods" section.
“The study extends previous findings by the same group. The authors should emphasize what new mechanistic insight is gained here.”
We have revised the manuscript text to emphasize new mechanistic insights, such as the role of promoter specificity in enhancing trans-splicing efficiency and the persistence of trans-splicing with minimal intron elements. These additions highlight the novel regulatory layers beyond our prior work, which was devoted to the study of regulatory sequences within intron 4.
“Figures are informative but dense. Label font size is small.”
We have increased the font size of the text in the figures to improve readability. Additionally, we have enhanced the contrast of the figures.
“Include specific concentrations and amounts of plasmid DNA used”
In the Materials and Methods section, we have specified the concentration of plasmid DNA as 200 ng/μL, which was used for embryo microinjections. In other sections, where concentrations differ from those recommended by the kit manufacturer, they are indicated.
“Authors should deposit plasmid maps, construct sequences in a repository (GEO).”
GEO does not accept plasmid sequences, only high-throughput data. NCBI Nucleotides accepts sequences of natural DNA. Therefore, we have included the sequences of the main plasmids in the supplementary materials.
Round 2
Reviewer 1 Report
Comments and Suggestions for Authors
The authors addressed most comments. To prove the linearity, the authors use a different way of explanation, but this is also acceptable.